# Effect of Providing Environmental Enrichment into Aviary House on the Welfare of Laying Hens

**DOI:** 10.3390/ani12091165

**Published:** 2022-05-02

**Authors:** Jiseon Son, Woo-Do Lee, Hee-Jin Kim, Bo-Seok Kang, Hwan-Ku Kang

**Affiliations:** Poultry Research Institute, Rural Development Administration National Institute of Animal Science, Pyeongchang 25342, Korea; wltjs1206@korea.kr (J.S.); woodo92@korea.kr (W.-D.L.); khj0175@korea.kr (H.-J.K.); kbs2901512@korea.kr (B.-S.K.)

**Keywords:** laying hen, aviary, feather pecking, enrichment, pumice stone, alfalfa hay, welfare

## Abstract

**Simple Summary:**

Behaviors such as feather pecking and cannibalism have resulted in several problems in the poultry industry. Owing to animal welfare concerns, the practice of beak trimming and use of cage systems have gradually reduced, and housing systems are currently being applied such as barn, floor, and aviary systems; however, the identification and control of feather peckers among many individuals is difficult in these systems. For this purpose, research is being conducted to observe the effect of improving the environment by providing various pecking materials. Our study investigated the environmental enhancing effects of providing pumice stone and alfalfa hay to laying hens in the aviary system. These materials were effective in improving egg production, maintaining health, and alleviating stress. The findings of this study could be useful in environmental management to improve the welfare of the poultry industry in the new rearing system.

**Abstract:**

This study aimed to determine the effects of providing environmental enrichment materials—pumice stone and alfalfa hay—to laying hens in the aviary system. A total of 2196 40-week-old Hy-Line Brown laying hens were randomly allotted to three treatment groups: (1) no enrichment (control; CON), (2) enrichment with pumice stone (PS), and (3) enrichment with alfalfa hay (HAY). Each treatment comprised four replicates of 183 hens each, and four of the same materials were provided per replicate. The experiment lasted for 26 weeks. Feed and water were provided ad libitum. As a result, the PS and HAY groups demonstrated increased egg production (*p* < 0.001). The HAY group showed a reduced rate of mislaid eggs (*p* < 0.01) and produced low egg weight and pale-yellow yolk (*p* < 0.05). Both enrichment materials decreased blood creatinine (CRE) or lactate dehydrogenase (LDH) in the blood and resulted in a significantly lower corticosterone (CORT) level (*p* < 0.05). However, the feather condition scores for the laying hens were similar across all treatments (*p* > 0.05). In summary, although pumice stone and alfalfa hay are effective in alleviating stress and improving the production of laying hens, additional environmental improvement studies are needed to contribute to reducing pecking behaviors in poultry farming.

## 1. Introduction

Pecking behaviors such as feather pecking, vent pecking, and cannibalism result in economic and welfare problems with regard to laying hens [1,2]. Studies have shown that cannibalism accounts for 17% of the total mortality in commercial housing systems, with feather pecking being observed in up to 80% of non-cage housing systems [3,4]. These behaviors are directly related to flock and group size, and worse feather condition and skin damage to body parts have been observed in larger groups [5]. Previously, conventional cage systems allowed for the easy observation of abnormal feather pecking behaviors in chicken coops, meaning it was an efficient method to remove the pecking individual, consequently reducing any damage caused by the behavior [6]. However, with a rising focus on animal welfare, the European Union (EU) initiated a ban on conventional caging for chickens in 2012. By 2014, 89.3% of poultry farms in Germany and 45% across the entire EU had converted to alternative, non-cage housing solutions [1,7]. Although these alternative housing systems such as aviaries and free-range systems are advantageous in improving the rearing environment of laying hens, increasing the awareness of egg consumers, and resulting in health benefits, they do result in greater numbers of pecking victims because identifying and removing pecking individuals is more challenging than in conventional cages [1,8]. A further challenge for controlling pecking is that beak trimming, a well-known and relied-upon method for reducing pecking occurrences, was banned in many countries from 2011 owing to animal welfare concerns (e.g., difficulty in eating, decreased activity, chronic pain). Therefore, there is an urgent need to find an alternative way to control pecking behavior without resorting to beak trimming [9,10,11,12].

In recent years, much research has been conducted on the design and management of systems for improving the welfare of chickens in poultry farming [13]. Among them, environmental enrichment has arisen as a potentially beneficial method for improving the natural behavior development of birds, while also reducing the frequency of abnormal and damaging behaviors, such as pecking [13]. In addition, environmental enrichment upgrades, such as the installation of nest boxes, perches, and straw and pecking devices, may alleviate negative emotional states, including fear and depression, and reduce feather-pecking behavior [14]. Studies have shown that native feather-pecking behavior is alleviated when the birds’ immediate environment is improved by adding stones, hay, and foraging material to the rearing facilities of laying hens [15,16]. However, items such as actuated feeders, shoelaces, and plastic rods did not raise awareness or interest in laying hens [14], and further research into pecking devices for commercial application is required to fully understand the types of activities and upgrades necessary for poultry enrichment [16]. Therefore, this study aimed to determine the effect of environmental enrichment materials, namely, pumice stone and alfalfa hay, on laying hens raised in the aviary system. As analysis items, egg production, egg quality, blood biochemical characteristics, serum corticosterone (CORT) levels, and feather condition were evaluated to elucidate the effect of these enrichment materials on improving the welfare of laying hens.

## 2. Materials and Methods

### 2.1. Experimental Animal and Design

For this experiment, a total of 2196 non-beak-trimmed 40-week-old Hy-Line Brown laying hens (*Gallus gallus domesticus*) were used. All birds were reared under the same conditions in a pullet rearing aviary, and after 16 weeks of age, all reared birds were transported to an adult aviary system. Each bird was randomly allotted to a selection of three treatments, with 732 birds per treatment. For each treatment, four replicates were conducted. The three experimental treatments were divided as follows: (1) no enrichment, i.e., the control (CON); (2) enrichment with pumice stone (PS); and (3) enrichment with alfalfa hay (HAY). Housing for the experiments included a modified two-tier commercial aviary system (Bolegg Terrace, Vencomatic Group, Eersel, The Netherlands), which was divided into pens with a floor area of 12.42 m^2^ (2.3 m × 5.4 m). The segmentation of the aviary system was modified to fit each pen. The first tier of the aviary was 100 cm and the second tier 220 cm above the floor. Both levels were equipped with round perches of metal measuring 35 mm in diameter, and nest boxes were situated in the first tier. The stocking density of the birds in the aviary system was lower than the established Korean regulations (17 birds/m^2^). No flocks had access to the outside. The light sources were LED lights (HATO Agricultural Lighting, Sittard, The Netherlands) placed in the ceiling on either side and strips of LED lighting near the chain feeder of the first tier and beneath the lower tier. The light intensity measured at head height was adjusted to about 30 lux using a lux meter (Testo AG, Lenzkirch, Germany), and the photoperiod was maintained at 16 h of light and 8 h of darkness (16L:8D). The temperature and humidity for the housing area were set at 22 ± 2 °C and 40 ± 5%, respectively. Adequate ventilation was maintained with a consistent fresh oxygen supply. The effect of environmental enrichment was analyzed for a total of 26 weeks, from 40 to 65 weeks of the hens’ age. The diet for the duration of the experiment consisted of a corn–soybean meal-based commercial diet in mash form, which is formulated to meet or exceed the nutritional requirements according to the age of the laying hens. Feed and water were provided for ad libitum consumption using a chain dispersal system.

### 2.2. Pecking Material

The pecking materials used were pumice stone and alfalfa hay (Figure 1A,B, respectively). Pumice stones from Mt. Baekdu were purchased from a bonsai shop (Wonju, Korea) and all stones were selected to be similar in size: approximately 24 cm in width, 17 cm in depth, and 20 cm in height. Alfalfa hay was purchased from Apet (Busan, Korea), with the sizing of the hay products measuring 30 cm wide, 21.5 cm deep, and 15 cm high. The pecking materials were considered at a ratio of 1:46 hens, with four materials distributed evenly in each pen. Each pecking material provided on the start date of the experiment was maintained without replacement during the experimental period.

### 2.3. Production and Mortality

The eggs laid by the birds in each experiment were collected once daily at 2 pm. The hen-day egg production ratio was calculated by dividing the total number of eggs collected by the number of live hens per day in each replicate. The collected eggs were classified into four types according to the method of Winkel et al. [17]: (i) first-class eggs (i.e., saleable eggs, normal eggs); (ii) second-class eggs (i.e., eggs with a cracked shell but intact membrane); (iii) third-class eggs (i.e., shell eggs); and (iv) mislaid eggs (i.e., those laid anywhere outside of the nest box). Mortality was recorded daily for the duration of the experiment.

### 2.4. Egg Quality

At the end of the experiment, among the normal eggs collected from each treatment group, 30 were randomly selected for quality analysis. The items analyzed included: egg weight, Haugh unit, albumen height, yolk color, eggshell color, eggshell thickness, and eggshell strength. Measurement of the egg weight, albumen height, Haugh unit, and yolk color was conducted using an egg multi-tester (EA-01 type, ORKA Food Technology Ltd., Ramat Hasharon, Israel). Eggshell color was measured using the eggshell color fan (Samyang Co. Ltd., Seoul, Korea) [18]. The strength of the eggshell was measured at the moment the shell broke using an eggshell force gauge (Egg Shell Force Gauge Model II, Robotmation Co., Ltd., Tokyo, Japan) by gradually increasing the pressure strength of the horizontally fixed egg to breaking point. The eggshell thickness was measured using a caliper (Digimatic Micrometer, Series 547-360, Mitutoyo, Japan) after the eggshell membrane was removed.

### 2.5. Blood Sampling and Analysis

At the end of the experiment, 40 hens (10 birds per pen) were randomly selected from each treatment and designated for blood sampling. The blood samples were collected in a serum separation tube (BD, Franklin Lakes, NJ, USA) from hens by wing venipuncture (wing vein). Following this, the serum was collected by centrifugation at 12,000× *g*, at 4 °C for 10 min, and stored at −80 °C until analysis. Serum biochemical variables included: aspartate aminotransferase (AST), alanine aminotransferase (ALT), lactate dehydrogenase (LDH), triglycerides (TG), total cholesterol (TCHO), glucose (GLU), total protein (TP), albumin (ALB), calcium (Ca), inorganic phosphate (IP), and creatinine (CRE). The serum biochemical variables concentration analysis was performed using an automatic chemistry analyzer (AU480 Chemistry Analyzer, Beckman Coulter Inc., Brea, CA, USA). Lastly, to analyze CORT (a stress hormone in the serum), a corticosterone enzyme-linked immunoassay kit (ADI-900-097, Enzo Life Science, Inc., Farmingdale, NY, USA) was used according to the protocols.

### 2.6. Feather Damage

The feather quality of hens was assessed using a scoring system that was animal-friendly, less stressful, and an effective method for determining feather damage. At the end of the experiment, 40 birds (10 birds per pen) were randomly selected for each treatment group, and the conditions of the head/neck, back, breast, wings, and tail were observed. The criteria of the score were based on Son et al.’s [19] method, with a score of 1–4 assigned to each part according to the classification of the welfare indicator. The classification according to score is as follows: (1) no damage (perfect feather); (2) slightly damaged (some feathers wrinkled or missing, no skin area denuded); (3) moderate damage (denuded area < 5 cm^2^ or 50% of area); and (4) very damaged (large denuded area > 5 cm^2^ or >50%). Photo examples of the score for each part are shown in Figure 2. The five scores were combined to give a total feather damage score ranging from 5 to 20 points for each hen.

### 2.7. Statistical Analysis

All data were analyzed in a completely randomized design with three treatments using the analysis of variance of a SAS 9.4 (SAS Institute, Cary, NC, USA). The housing pen (183 laying hens each) was the experimental unit for the analysis of performance parameters. The data were pooled for all experiment periods and the results are outlined in the following section. The egg quality items were analyzed statistically at the end of the last experimental week by considering the number of eggs as the experimental unit. Individual birds served as the experimental unit for blood variables, as well as for the feather damage score. Duncan’s multiple range test was carried out to assess the significant differences for all analyzed variables at the probability level of *p* < 0.05 among the treatments. The data are expressed as means and the standard error of the means.

## 3. Results

### 3.1. Laying Hen Performance

Table 1 shows the differences in mortality, egg production, and egg external quality according to each tested environmental enrichment. It was noted that the enrichment materials did not affect mortality. However, elevated egg production levels were found in the PS and HAY groups compared to CON (*p* < 0.001). In the case of egg external quality, when alfalfa hay was given as a pecking material, the first-class egg production rate was the highest (*p* < 0.05), and the proportion of mislaid eggs was the lowest (*p* < 0.01).

### 3.2. Egg Quality

The effects of environmental enrichment materials on egg weight, Haugh unit, albumen height, eggshell color, egg yolk color, eggshell strength, and eggshell thickness are summarized in Table 2. The PS treatment showed the effect of improving the weight of eggs, while HAY treatment produced a pale-yellow color egg yolk (*p* < 0.05). No differences were observed in terms of Haugh unit, albumen height, eggshell color, eggshell strength, or eggshell thickness.

### 3.3. Blood Variables

The data from the blood biochemical variables taken at the end of the experiment are presented in Table 3. The provision of alfalfa hay significantly reduced CRE in laying hens and showed a lower LDH level together with the PS group (*p* < 0.05). Other variables showed no significant difference.

### 3.4. Serum Corticosterone

Table 4 shows the changes in CORT according to the pecking materials provided to laying hens in the aviary system. The groups treated with pecking materials (PS and HAY) showed significantly lower CORT levels than CON (*p* < 0.05).

### 3.5. Feather Score

Table 5 summarizes the results of the analysis of the environmental enrichment effect of the pecking materials by observing the state of the feathers for each part of the laying hen. The enrichment groups (PS and HAY) did not show any effect on improving the feather condition of laying hens (*p* > 0.05).

## 4. Discussion

This study was conducted to confirm the environmental enrichment effects of PS or HAY treatment on laying hens in the aviary system. Although feather pecking is a natural part of a hen’s behavior, it is considered abnormal when consistently used against other birds. This behavior can develop into severe pecking, where there is a risk that it can damage the skin of other birds, and in some cases develop into cannibalism [20].

Severe feather pecking is considered a serious economic and welfare problem for the laying hen industry worldwide, as it reduces the egg production of flocks and can include weight loss, increased mortality, increased feed consumption, and decreased feed conversion rates [20,21,22]. According to several reports, a large proportion of the laying hens that die do so as a result of feather pecking and cannibalism [23,24]. In particular, the aviary system, where it is difficult to select a feather pecker among various rearing facilities, showed a higher mortality rate (1% or more per week) [1,25,26,27]. One particular cause that has been associated with feather-pecking behavioral traits is the inhibition of foraging behavior such as ground pecking or a lack of environmental stimuli. Studies conducted on foraging behaviors have revealed that it is closely linked to limiting environmental conditions, most commonly resulting in pecking naturally at the feathers of other hens. Much research has since been conducted on the effect of providing enrichment upgrades and materials to laying hen housing to reduce the occurrence of severe feather pecking [28]. Tainika et al. [29] found that providing litter to laying hens reduced the number of pecking behaviors and mortality, and Gvaryahu et al. [30] found that provision of environmental enrichment materials significantly reduced mortality when compared to control groups. It was also reported that nest boxes and abrasive strips in cages contribute to reducing mortality in laying hens [31]. In particular, it was found that when pecking stones were provided to laying hens, their beak shortened and the sharpness reduced [32]. This affects feather-pecking behavior and has been found to be associated with mortality and improved feather condition [32,33]. However, contrary to this finding, other studies noted that some environmental enrichment materials, such as pecking stones and strings, whole oats, and litter, did not have significant effects on feed conversion ratio, feed intake, body weight, egg production, proportion of floor eggs, or mortality [28,33,34]. In this study, when pumice stone and alfalfa hay were provided as enrichment materials, it did not affect the mortality rate as in some studies; however, it did contribute to improving egg production and the proportion of mislaid eggs.

Haugh unit and eggshell color are traits that determine the quality of eggs and affect consumer preference [35]. Additionally, there are many factors that affect egg quality, such as heredity, age, and feed, and recently, research on the effect of quality improvement through environmental enrichment in rearing facilities has been conducted [29]. In our study, the HAY group influenced egg weight and yolk color, whereas PS did not show any significant difference between the egg quality traits. Bari et al. [36] showed that when enrichment such as balls, brooms, buckets, ropes, and an H-shaped perch were provided, eggshell quality and yolk color were improved compared to the control, meaning environmental enrichment directly affects egg quality. Hu et al. [37] reported that eggshell quality was improved when the environment was enhanced with water chilled perches. Furthermore, Galic et al. [38] reported that free-range hens improved the ratio of yolk to albumen compared to hens raised in cages. In a study by Schreiter et al. [39], the use of alfalfa hay and pecking stone as pecking devices in Lohmann Selected Leghorn and Lohmann Brown laying hens was effective in improving egg weight, but not in albumen consistency and eggshell stability. In particular, Johannson et al. [40] reported that when barley silage was provided to laying hens in group-housed enriched cages to reduce pecking, the yolk color was darker due to the ingestion of the barley silage. In this study, it is thought that that the significant effect of egg yolk color in the HAY group was due to ingestion of alfalfa hay during the experiment.

In veterinary medicine, conducting blood variables analyses is important for objectively evaluating the health of livestock and for accurately diagnosing diseases [41,42]. Factors that affect blood components include season, age, rearing, nutrition, and stress, and the applied environmental conditions also affect animals [43,44]. According to several reports, the selection of optimal environmental conditions is critical because the various rearing environments of poultry greatly affect many components in the blood, i.e., AST, TG, ALB, Ca, IP, creatine kinase (CK), and the ratio of heterophils and lymphocytes (H/L) [24,45]. In our study, the provision of pumice stone and alfalfa hay has been shown to have a positive effect on the CRE and LDH in the blood of laying hens. LDH is an enzyme that is increased due to damage or stress in the liver cells, and CRE is an indicator of kidney dysfunction [46,47,48]. Therefore, it appears that the substances provided in this study had a positive effect on the stress alleviation and health of the laying hens. Nicol et al., (2011) found that enriching laying hens with a variety of novel objects (balls, green apples, balloons, plastic bottles, aluminum cans) had a positive effect on the H/L ratio, an indicator of stress in the body, and lowered blood sugar levels [49]. Yildirim and Taskin [44] reported that when Ross 308 broilers were environmentally enhanced with a mirror, ball, perch, or dust bath, the ball group had significantly lower hematocrit (HCT) and platelet count (PLT), and all environmental enrichment groups had lower white blood cells (WBCs) than the control group.

Corticosterone levels in poultry can be analyzed using a variety of samples including feathers, feces, and blood. Feather corticosterone (fCORT) assays are used to assess long-term stress in birds as they provide the ability to assess the adrenocortical activity during the growth period [50]. Although this method is also useful in terms of animal welfare analysis, many potential factors are known to influence fCORT [50,51,52]. Studies have shown that fCORT can be reduced by physical damage and varies with season, time, and location, with significant differences due to fecal contamination [50,53]. It is also reported that further investigation is necessary to standardize corticosterone in feathers because it significantly changes according to the treatment method of feathers, and the concentration also differs according to the laying hen species and the sampling parts within the individual [51,52]. The analysis of corticosterone in feces is also a useful method for assessing animal welfare and has the advantage of readily available samples [54]. However, there are several factors that affect the concentration in this assay, and there is a caution that a fecal sample must be collected within 30 min after excretion to analyze the exact concentration [54,55].

Corticosterone level analysis using blood is a method that has been used for a long time and is being used as an indicator of animal welfare in many studies [53,56,57,58]. High CORT concentration is associated with hunger, heat, fear, stress, and the poor environment of the laying hens [31]. This hormone is a stress hormone that interacts with serotonin (5-HT) and is influenced by potentially threatening stimuli in the laying hen, such as levels of feather pecking behavior, namely, high feather-pecking line and low feather-pecking line [31,59,60]. Accordingly, many researchers have analyzed changes in stress hormones by providing materials to the poultry-rearing facilities, and several materials have been shown to have a stress-reducing effect [61,62]. In Fairhurst et al.’s [63] study, when the environment for laying hens was improved by providing toys, the concentration of CORT increased immediately after adding the novelty objects, but the concentration decreased compared to the control over time. In contrast, when the environment was improved with plastic boxes and hay bales, there was no significant effect on CORT and 5-HT concentrations in the laying hens [62], with environmental improvements with nesting boxes also having no significant effect [64]. In this experiment, the provision of enrichment materials (pumice stone or alfalfa hay) was judged to be effective in reducing stress in laying hens by showing significantly lower CORT levels compared to the control group.

Feather damage is closely related to feather pecking behavior and is considered an indicator for evaluating the animal welfare of laying hens. Several recent studies have investigated the effects of providing novelty objects and adjusting various housing systems to reduce the damage from feather-pecking behavior [4,65]. Many researchers have reported that the provision of enrichment materials and the stimulation of foraging behavior can influence the occurrence of feather-pecking behavior. The provision of pecking stones, pecking blocks, and lucerne bales to hens in the aviary system significantly reduced feather pecking and aggressive pecking behavior [4]. When plastic boxes or hay bales were randomly applied to floor pens, hay bales appeared to be a promising environmental enrichment material for mitigating feather pecking among hens [62]. It was also shown that toys and gravel stones were suitable for mitigating feather pecking when pecking stones, gravel, oyster shells, grains scattered in the litter, and toys were applied to laying hens in a multi-tiered aviary system [66]. In particular, the introduction of pecking devices to improve the welfare of the laying hens in the cage reduced agonistic behaviors and promoted some instinctive behaviors of the hens [16]. However, unlike the above studies, the pumice stones and alfalfa hay provided in this study did not significantly affect the feather scores of the laying hens.

## 5. Conclusions

The results from this study effectively illustrate the positive effect of pumice stone and alfalfa hay, improving the welfare environment of laying hens in the aviary system. The provided materials directly correlated to improved egg production, and also to reduced incidences of mislaid eggs. The provision of enrichment materials has also been shown to significantly lower the levels of CRE, LDH, and CORT in the blood, helping to improve hen health as well as to relieve stress. The provision of pumice stone or alfalfa hay is believed to have positive effects on laying hens and improve the welfare of poultry-rearing facilities. However, to solve the mortality and economic problems caused by feather pecking and cannibalism, further research is required to understand how to improve the rearing environment for hens in addition to searching for environmental enrichment materials.

## Figures and Tables

**Figure 1 animals-12-01165-f001:**
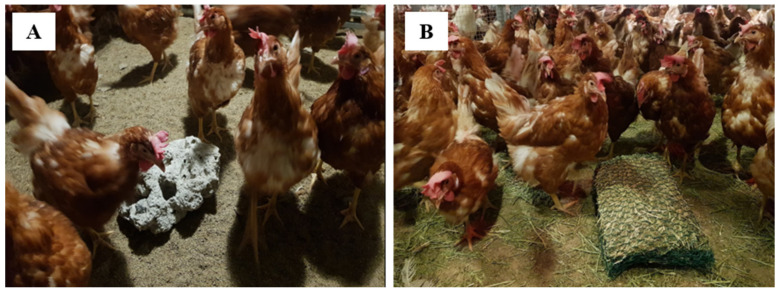
Environmental enrichment through different types of pecking materials. (**A**) Pumice stone; (**B**) alfalfa hay.

**Figure 2 animals-12-01165-f002:**
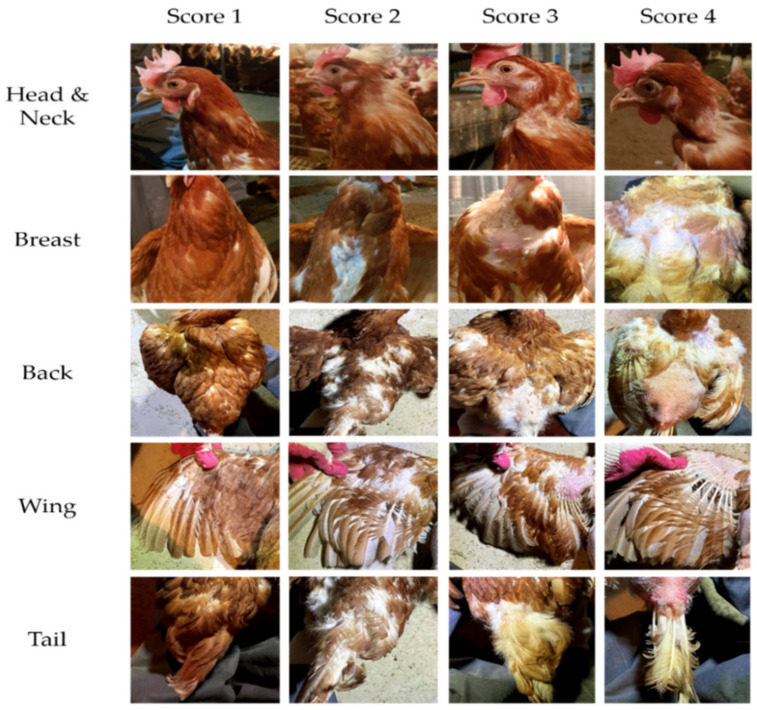
Score based on feather quality for each part of the laying hen.

**Table 1 animals-12-01165-t001:** Means of mortality, egg production, and egg external quality indicators according to environmental enrichment materials.

Item	Treatment ^1^			SEM ^2^	*p* Value
	CON	PS	HAY		
Mortality/laying period, %	1.040	0.862	0.846	0.025	0.694
Egg production, %	78.34 ^b^	81.11 ^a^	80.73 ^a^	0.427	<0.001
Egg external quality					
First-class eggs, % ^3^	98.82 ^b^	98.84 ^b^	99.07 ^a^	0.038	0.017
Second-class eggs, % ^4^	0.068	0.063	0.088	0.007	0.203
Third-class eggs, % ^5^	0.016	0.018	0.032	0.003	0.116
Mislaid eggs, % ^6^	1.094 ^a^	1.075 ^a^	0.815 ^b^	0.039	0.007

^1^ CON: no enrichment; PS: enrichment with pumice stone; HAY: enrichment with alfalfa hay. ^2^ SEM, standard error of the mean. ^3^ First-class eggs: saleable eggs, normal eggs. ^4^ Second-class eggs: eggs with a cracked shell but intact membrane. ^5^ Third-class eggs: shell-eggs. ^6^ Mislaid eggs: eggs laid outside the nest. ^a,b^ Means within a same row with different letters differ significantly at *p* < 0.05.

**Table 2 animals-12-01165-t002:** Means of internal egg quality indicators of laying hens according to environmental enrichment materials.

Item	Treatment ^1^			SEM ^2^	*p* Value
	CON	PS	HAY		
Egg weight, g	63.2 ^ab^	64.3 ^a^	61.0 ^b^	0.455	0.010
Haugh unit	77.9	76.1	74.9	1.269	0.637
Albumen height, mm	6.43	6.22	6.07	0.139	0.575
Egg yolk color	8.30 ^a^	8.40 ^a^	7.50 ^b^	0.135	0.010
Eggshell color	11.0	11.3	11.2	0.131	0.573
Eggshell strength, kg/cm^2^	3.58	3.24	3.55	0.091	0.235
Eggshell thickness, mm	0.426	0.418	0.429	0.003	0.316

^1^ CON: no enrichment; PS: enrichment with pumice stone; HAY: enrichment with alfalfa hay. ^2^ SEM, standard error of the mean. ^a,b^ Means within a same row with different letters differ significantly at *p* < 0.05.

**Table 3 animals-12-01165-t003:** Means of blood variables of laying hens according to environmental enrichment materials.

Item	Treatment ^1^			SEM ^2^	*p* Value
	CON	PS	HAY		
AST ^3^, U/L	225	239	251	9.18	0.503
ALT ^4^, U/L	5.01	3.33	3.62	0.328	0.081
LDH ^5^, mg/dL	1331 ^a^	1033 ^b^	1035 ^b^	55.2	0.038
TG ^6^, mg/dL	1122	1044	1153	33.1	0.393
TCHO ^7^, mg/dL	140	137	125	5.00	0.404
GLU ^8^, mg/dL	96.4	113.3	117.8	5.19	0.207
TP ^9^, g/dL	5.92	5.81	5.75	0.05	0.392
ALB ^10^, g/dL	2.16	2.12	2.13	0.013	0.394
Ca ^11^, mg/dL	26.3	25.0	26.4	0.341	0.170
IP ^12^, mg/dL	5.91	5.66	5.81	0.112	0.675
CRE ^13^, mg/dL	0.28 ^a^	0.28 ^a^	0.26 ^b^	0.003	0.035

^1^ CON: no enrichment; PS: enrichment with pumice stone; HAY: enrichment with alfalfa hay. ^2^ SEM, standard error of the mean. ^3^ AST, aspartate aminotransferase. ^4^ ALT, alanine aminotransferase. ^5^ LDH, lactate dehydrogenase. ^6^ TG, triglycerides. ^7^ TCHO, total cholesterol. ^8^ GLU, glucose. ^9^ TP, total protein. ^10^ ALB, albumin. ^11^ Ca, calcium. ^12^ IP, inorganic phosphate. ^13^ CRE, creatinine. ^a,b^ Means within a same row with different letters differ significantly at *p* < 0.05.

**Table 4 animals-12-01165-t004:** Means of serum corticosterone concentration in laying hens according to environmental enrichment materials.

Item	Treatment ^1^			SEM ^2^	*p* Value
	CON	PS	HAY		
CORT ^3^, pg/mL	218.6 ^a^	174.6 ^b^	163.2 ^b^	9.301	0.033

^1^ CON: no enrichment; PS: enrichment with pumice stone; HAY: enrichment with alfalfa hay. ^2^ SEM, standard error of the mean. ^3^ CORT, corticosterone. ^a,b^ Means within a same row with different letters differ significantly at *p* < 0.05.

**Table 5 animals-12-01165-t005:** Means of feather scores of each part of laying hens’ body according to environmental enrichment materials.

Item	Treatment ^1^			SEM ^2^	*p* Value
	CON	PS	HAY		
Head	2.175	1.825	2.025	0.067	0.966
Back	3.575	3.525	3.575	0.078	0.956
Breast	3.000	2.950	3.535	0.105	0.053
Wing	2.675	2.625	2.725	0.094	0.912
Tail	3.400	3.150	3.350	0.089	0.483
Total	14.88	14.08	15.10	0.357	0.471

^1^ CON: no enrichment; PS: enrichment with pumice stone; HAY: enrichment with alfalfa hay. ^2^ SEM, standard error of the mean.

## Data Availability

The data are available on request from the corresponding author.

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
