# Peer review of "Effect of Providing Environmental Enrichment into Aviary House on the Welfare of Laying Hens"

_animals, 2022, doi:10.3390/ani12091165_

Round 1

Reviewer 1 Report

Animals 2022

Manuscript ID: animals-1672913

Title: Effect of providing environmental enrichment of materials into aviary house of laying hens

General comments:

In this manuscript the authors addressed an important and well-studied subject, analysing the effects of two materials to promote environmental enrichment in aviaries on the welfare of laying hens. Despite the approach is not totally innovative, the manuscript offers important practical results, showing an easy and cheap way to improve the welfare of lying hens kept in aviaries. The manuscript is well written and interesting to read. Additional comments and suggestions are presented below.

Additional comments

Title

I recommend reviewing the title, suggestion: “Effect of providing environmental enrichment into aviary house on the welfare of laying hens”

Introduction

L64. Replace "native" for “negative".

Material and methods

L89. Insert "of hens age" after “weeks".

L95. Insert "A and B, respectively)" after "Figure 1”.

L132. Consider replacing "parameters" for “variables".

L136. Consider replacing “parameters” for “variables”.

L162. Consider replacing “parameters” for “variables”.

L164. Consider replacing “parameters” for “variables”.

Results

Table 1.  Suggestion: Replace Table 1 caption for "Means of mortality, egg productivity and egg external quality indicators according to environmental enrichment materials".

Table 1. Mortality % - Very high mortality, can you explain that?

Table 2. Suggestion: Replace Table 2 caption for "Means of internal egg quality indicators of laying hens according to environmental enrichment materials”.

L191. Consider replacing "parameters" for “variables".

L192. Consider replacing "parameters" for “variables".

L194. Consider replacing "parameters" for “variables".

Table 3. Suggestion: Replace Table 3 caption for "Means of blood variables of lying hens according to environmental enrichment materials”.

Table 4. Suggestion: Replace Table 4 caption for "Means of plasma corticosterone concentration in laying hens according to environmental enrichment materials”.

Table 5. Suggestion: Replace Table 5 caption for "Means of feather scores of each part of lying hens body according to environmental enrichment materials”.

Discussion

L224-225. After reading the two articles cited here, I noticed that these information are not completely in line with their results. Check!

L237. Insert "," after “materials".

L243. Consider replacing "parameters" for “traits".

L248. Consider replacing "parameters" for “traits".

L264. Consider replacing "parameters" for “variables".

L264-177. Please, make clear what are the relationships between your results and those related with the articles here cited.

L278. By whom? Please, insert citations.

Reviewer 2 Report

The authors have addressed a very important and still topical issue of environmental enrichment of laying hens. The authors investigated the effects of pumice and alfalfa hay as enrichment material on blood parameters, plumage condition and performance.

However, I do have a few comments below regarding the design of the study in particular, as well as the discussion that follows.

The laying hens for the study were housed very late at 40 weeks, please explain briefly what was the reason for this. It is already common practice in many countries to house the laying hens much earlier, before the start of laying (around 18/19 weeks of age) to give them a period of acclimatization to the new housing system before the start of laying.

Please also explain the conditions during rearing and therefore also the origin of the animals, as this often already has an impact on later behavior and any problems that occur as a result from this. The importance of the rearing phase needs to be discussed in the context of feather pecking and cannibalism.

There is a lack of  a more detailed information on the lighting and the illuminants used, as well as information on the light spectrum and freedom from flickering. Was there access to daylight? If the light is not flicker-free or sunlight enters the coop at awkward angles, this can also trigger severe feather pecking and cannibalism.

According to the material and method section, the pumice stones were taken directly from a mountain. Were they cleaned or  treated otherwise before they were offered to the hens? If not, how is this to be assessed in terms of animal hygiene and consumer protection.

Pumice stones, when used extensively by hens, can help to naturally smoothen the beaks of hens, allowing them to have a rounded beak similar to that of beak-trimmed birds. This aspect is missing from the discussion. Have the beaks of the pumice group been examined and the lengths or beak condition measured?

Corticosterone levels from blood serum can be affected by animal handling alone, as this can stress the animals as the animals are caught and restrained. Better detection methods from feathers and feces already exist. These aspects are missing from the discussion.

Mortality rates are extremely high in all three groups. Causes for this are not mentioned. It is also not discussed further. Animal health examinations other than blood parameters are not presented. What was the vaccination status of the hens? Were deceased hens pathologically examined and samples taken? No  termination criteria for the experiment are mentioned. For ethical reasons, I am concerned whether the animal welfare standard was met here and whether the animals experienced pain, suffering or harm during the study.

Reviewer 3 Report

This is a useful paper about a subject that is still very much in the attention of the poultry industry.

A few remarks:

  • It was decided to group all subjects within each treatment group together, even though there were 4 repeats. This is probably justified, but it would be helpful to analyse significance of differences between repeats within treatment groups. If such an analysis shows there are no differences between treatment groups, then grouping them all together is justified. It may not be necessary to present the analysis extensively in this paper, but a remark could be made that such an analysis was carried out and that it showed that the results of the four repeats could be grouped at treatment level.
  • Was there a particular reason why 40 week old hens were entered in the experiment? Wouldn't it have been more logical to enter the hens at approximate age 26 weeks, at start of laying, as they already have a life behind them when entering at 40 weeks? It would be helpful if the rationale would be explained.
  • The paper states that the stocking density as it was regulated in Korea was used. It would be helpful to mention what that stocking density is, in number of birds met square meter, or number of kilogrammes of bird per square meter.
  • in the tables, it may be useful to indicate for each value how they differ significantly from the other values in the table, for example by putting a superscript letter (a, or b, depending on significant differences) next to each value if they are significantly different from other values in the same line in the table. for example, it seems that egg production is higher in the treatment groups as compared to the control group, but that the vakues of both treatment groups are not significantly different. by giving the control group a superscript a and both treatment groups a superscript b, that can be made clear.
